# Dehumanization as a Response to Uncivil and Immoral Behaviors

**Laura Rodríguez-Gómez** [ID], **Naira Delgado** [ID], **Armando Rodríguez-Pérez** *[ID], **Ramón Rodríguez-Torres** [ID]
**and Verónica Betancor** [ID]

Department of Cognitive, Social and Organizational Psychology, Faculty of Psychology, University of La Laguna, 38200 San Cristóbal de La Laguna, Santa Cruz de Tenerife, Spain
* Correspondence: arguez@ull.es

**Abstract:** Theoretical approaches to dehumanization consider civility to be an attribute of human uniqueness (HU). However, studies that explore the links between civility and humanness are scarce. More precisely, the present research tests whether there is a consistent relationship between civility and HU. Method and results: The first study ($N$ = 192; $M_{age}$ = 19.91; $SD$ = 2.70; 69% women) shows that individuals infer more HU traits in the agents of civil behaviors compared to agents of other positive behaviors that are not related to civility. The second study ($N$ = 328; $M_{age}$ = 19.69; $SD$ = 3.65; 77% women) reveals that uncivil and immoral behaviors displayed a similar pattern of inference of HU traits; however, moral behaviors were more associated with human nature than civil behaviors. Conclusions: Overall, results confirmed that civil behaviors facilitate the inference of humanness, specifically of HU traits, and that civil and moral behaviors are not equivalent in terms of the human inferences to which they lead.

**Keywords:** civility; dehumanization; human uniqueness; animalistic dehumanization; morality

## 1. Introduction

Social norms are essential sources of information that people use to orient themselves in the complex social world. Among social norms, research points out the importance of civility in human interactions [1–4]. When we talk about civility, we refer to a type of ethical behavior that includes courtesy, manners, and good citizenship. However, experts today argue that civility encompasses more than good manners and etiquette. It also requires an awareness that extends beyond the self, involving respect and concern for the well-being of others and of the community [5,6]. Therefore, along with good manners, civil behavior involves tolerance, self-restraint, commitment to others, social involvement, responsibility, and an active engagement in creating, protecting, and sustaining the community [7].

In the last few decades, the social sciences have recently focused on its opposite, incivility [8,9]. In fact, uncivil behavior increasingly appears as the subject of daily conversation and media coverage because it involves an attack on the community's social norms. Unlike criminal acts, uncivil behaviors are not serious or dangerous enough to merit police attention or be a reason for systematic repression. In addition, on many occasions, they indicate behavior with an unknown intent to damage other citizens [10]. However, they do negatively affect people and often threaten those who are affected [11]. In this sense, uncivil behaviors have been evaluated as one of the primary urban factors that produce the most stress among citizens and that most reduce their quality of life within the context of the community [12,13]. Incivility thus becomes a negative element of social human behavior that clearly affects public health; it must be understood in order to create effective interventions to eradicate it.

The two studies presented below examine the links between uncivil behavior and the denial of full human potential to be considered an individual, that is, dehumanization. More precisely, the aim of the first study is to verify whether civility is related to one of



the specific components of humanness, human uniqueness (HU). In the second study, we extend the analysis of the link between civility and human uniqueness to morality, which constitutes a theoretical advancement in the understanding of the constructs of civility and humanness.

### 1.1. The Relationship between Humanness and Civility

Most of the measures of dehumanization have indirectly shown a possible link between civility and humanness [14–16]. Probably the most direct link between civility and humanness appears in the dual model of dehumanization [17], which defines humanness in two ways: via essential attributes that do not distinguish humans from other creatures (but that constitute humans' natural attributes) or as attributes exclusive to humans (compared to other species). The former comprises attributes of human nature (HN), including emotions, warmth, open-mindedness, agency, and the capacity for depth. Therefore, dehumanizing a group by depriving it of these attributes would equate to turning its members into automata (mechanistic dehumanization). The latter comprises attributes of human uniqueness (HU), which include civility, refinement, moral sensitivity, rationality, and maturity. Therefore, dehumanizing a group by depriving it of these attributes would equate to turning its members into animals (animalistic dehumanization).

The distinction between these two types of dehumanization has been the subject of much research [18]. However, within the framework of this theory, there is a general lack of empirical studies showing that the characteristics thought to be associated with HU are indeed empirically associated with this sense of humanness. Specifically, no exploration has been made of the extent to which people perceive a relationship between civil behaviors and humanness, and, consequently, the extent to which they make dehumanizing inferences about those who exhibit uncivil behaviors. In this sense, exploring the link between civility and HU traits constitutes a theoretical advancement in the understanding of the construct of dehumanization, especially in the theoretical framework of the dual model of dehumanization.

### 1.2. Moral Behaviors, Civil Behaviors, and Humanness

Previous research has found that people perceive morality as distinctively human, with immorality representing a lack of full humanness. For example, the relationship between immoral behavior and dehumanization appears in several studies by Bastian et al. [19,20], in which agents who performed harmful behaviors often faced dehumanization. Other investigations found that targets who were perceived as lacking moral qualities (e.g., low levels of honesty, sincerity, or trustworthiness) were attributed fewer human traits than were highly moral targets.

According to the dual model of dehumanization [17], morality and civility are both a feature of HU. Haslam [17] argued that the skills necessary to demonstrate competence (rationality and maturity) and to be moral (moral sensibility) are both high-order cognitions exclusive to human beings, that is, HU traits. In line with this reasoning, it can be expected that there will be no differences in the attribution of HU traits to targets performing both civil and moral behaviors. The present research sought to extend prior research on the dehumanized perception of perpetrators of immoral behaviors by testing whether uncivil behaviors may produce a similar effect upon the ascription of humanness.

### 1.3. The Present Research

In spite of the theoretical relevance of the concept of civility in the study of dehumanization, there has been little research into civil behavior or how it is associated with humanness. A recent study by Rodríguez-Gómez et al. [21] concluded that both uncivil and civil behaviors are implicitly associated with human concepts. However, to our knowledge, there is no evidence of the role of civil and uncivil behaviors in the inference of uniquely human traits. Since previous studies showed that the way people may be subtly ascribed or denied humanness has implications for judgments toward blame or punishment [19],

no study has tested the relationship regarding the inference of humanness when viewing others doing (un)civil and (im)moral acts.

The purpose of the present research is to clarify the association between civility and humanness. Specifically, it seeks to test out the hypothesis that civility is related to one of the specific dimensions of humanness: human uniqueness.

In the first study, we compare the trait inferences displayed toward the targets of civility versus other positive behaviors, and incivility versus other negative behaviors. The second study further analyses how civility and HU link to morality. Specifically, we compare the trait inferences that are displayed toward the targets of civil, moral, and other positive behaviors, and the trait inferences that are displayed toward the targets of uncivil, immoral, and other negative behaviors. All data and study materials are available for download at https://osf.io/jfg6a/?view_only=6a1e6b210338480e9eb383d191c80a76 (accessed on 12 September 2022).

### Study 1

The present study aims to explore whether civil behaviors could more often lead to the attribution of HU traits compared to other positive behaviors unrelated to civility. Simultaneously, we explore whether uncivil behaviors (through a lower attribution of HU traits compared to other negative behaviors unrelated to incivility) more often cause animalistic dehumanization compared to these other behaviors, as predicted by Haslam's theory [17]. Therefore, we expect: a greater inference of traits with high HU–low HN compared to other traits for civil behaviors (Hypothesis 1—H1), a lower inference of traits with high HU–low HN compared to other traits for uncivil behaviors (Hypothesis 2—H2), a greater inference of traits with high HU–low HN for civil behaviors compared to positive behaviors (Hypothesis 3—H3), and a lower inference of traits with high HU–low HN for uncivil behaviors compared to negative behaviors (Hypothesis 4—H4).

## 2. Materials and Methods

### 2.1. Participants

A total of 192 university students (132 female), all residents in Spain, participated in this study ($n$ = 47 in civil, $n$ = 49 in uncivil, $n$ = 46 in positive neutral, and $n$ = 50 in negative neutral conditions). The average age was 19.91 ($SD$ = 2.70) and ranged from 18 to 43 years old. A sensitivity analysis conducted with G*Power [22] revealed that the sample was sufficient to detect small effects of $f$ = 0.10 (equivalent to partial $\eta^2_p$ = 0.01) assuming an alpha coefficient of 0.01 and power of 0.95 (mean correlation among repeated measures = 0.83).

### 2.2. Instruments

*The behaviors.* Two civil behaviors and two uncivil behaviors were selected from a pretest study of civil and uncivil behaviors ($N$ = 360; $n$ = 261 female participants; $M_{age}$ = 20.01; $SD$ = 3.46). Specifically, the two civil behaviors were: "Think about the people who pick up after their dog after it has done its business when out on a walk" and "Think about the people who deposit their glass bottles in the glass recycling bins". The two uncivil behaviors were: "Think about the people who don't use the bike lane" and "Think about the people who leave their garbage out on the street instead of placing it in the bin". In addition, four behaviors unrelated to civility were included in the study. These two positive and two negative behaviors had been identified in another pre-test study ($N$ = 64; $n$ = 52 female participants; $M_{age}$ = 20.50; $SD$ = 3.91). The positive behaviors were "Think about the people who do their grocery shopping" and "Think about the people who tend to go for walks". The negative behaviors were "Think about the people who give their uninformed opinion about anything" and "Think about the people who waste their time instead of making the most of it". The criteria for selecting civil and uncivil behaviors required they represent high versus low civility and simultaneously be comparable in valence to the behaviors unrelated to civility. We compared the means of civil and uncivil behaviors to ensure

differences in civility and valence for each type of behavior. Another analysis of means verified that the civil behaviors were different from the neutral positive behaviors in civility but not in valence. The same analysis was performed for the uncivil behaviors and for the neutral negative behaviors and confirmed the differences in civility but not in valence (see the Supplementary Materials for details of the analysis).

*The traits.* The traits presented were all positive and represented the four groups defined by Haslam and Bain [23], resulting from crossing the HN and HU dimensions. The traits were selected from a pretest study that included 144 traits with scores for HN, HU, and valence ($N = 100$; $n = 70$ females; $n = 30$ males; $M_{age} = 20.12$; $SD = 3.48$). The 16 traits selected included the high HN–high HU traits of *passionate*, *idealistic*, *imaginative*, and *rational*; the high HN–low HU traits of *active*, *curious*, *efficient*, and *emotional*; the low HN–high HU traits of *cultured*, *humble*, *tolerant*, and *refined*; and the low HN–low HU traits of *uninterested*, *relaxed*, *satisfied*, and *serene* (see the Supplementary Materials for all analyses).

*2.3. Procedure*

Participants completed one of four versions of a paper-and-pencil questionnaire that asked them to form an impression of the type of people who exhibit certain behaviors and then to respond to a list of traits. Each questionnaire described two behaviors. For the civil condition, the questionnaire included two episodes of civil behaviors. For the uncivil condition, the questionnaire described two episodes of uncivil behaviors. Two control conditions contained two episodes of positive (vs. negative) behaviors that were unrelated to civility.

After reading each description, the participants received a list of 32 traits. Specifically, they were asked to score each trait according to the image they had formed of the person exhibiting the behaviors, on a scale from 1 (*The trait does not describe at all the type of person I imagined*) to 5 (*The trait fully describes the type of person I had imagined*). Of the 32 traits listed, 16 corresponded to the four types of traits defined by Haslam and Bain [23], and the other 16 were fillers. The experimental traits were mixed with the fillers, and the list was presented in two different orders. Half of the participants responded to one list, and the other half responded to the same list presented in reverse order.

*2.4. Data Analysis*

IBM SPSS Statistics (Version 25) was used for the analyses, with a significance level set at 0.05. After conducting an analysis of variance (ANOVA) of $2 \times 2 \times 2 \times 2$, several post hoc Bonferroni-corrected tests were conducted with the four types of traits derived via crossing high versus low HU traits and high versus low HN traits. First, the scores on the four traits in each behavior were compared to test H1 and H2. Second, the four behaviors in each trait were compared to test H3 and H4.

**3. Results**

To verify whether HU traits are more characteristic than other traits for civil behaviors and less characteristic than other traits for uncivil behaviors, a 2 (type of behavior: civility vs. neutral) $\times$ 2 (valence of behavior: positive vs. negative) $\times$ 2 (HN traits: high vs. low) $\times$ 2 (HU traits: high vs. low) ANOVA was carried out, with the behavior type and behavior valence as between-subjects variables and HN and HU as within-subjects variables. The differences in the means of the traits that scored high and low in HN and HU for each type of behavior are shown in Supplementary Table S4.

The ANOVA showed that the four-way interaction was statistically significant, $F_{(1,188)} = 3.96$, $p = 0.048$, $\eta^2_p = 0.02$ (the ANOVA results are summarized in Supplementary Table S5). To test our hypothesis, we made several post hoc Bonferroni-corrected tests between traits for each behavior. First, the results for civil behaviors revealed that, in line with H1, the traits for high HU–low HN were more characteristic than were the other traits ($p < 0.001$, Cohen's $d = 0.64$ for high HU–high HN; $p = 0.037$, $d = 0.25$ for low HU–high HN; $p < 0.001$, $d = 96$ for low HU–low HN).

Second, the results also confirmed H2's expected lower inference of traits that were high HU–low HN compared to other traits for uncivil behaviors. Specifically, the post hoc Bonferroni-corrected tests showed that traits for high HU–low HN were less characteristic than were other traits ($p < 0.001$, $d = 0.70$ for high HU–high HN; $p < 0.001$, $d = 0.83$ for low HU–high HN; $p < 0.001$, $d = 1.51$ for low HU–low HN).

This pattern of responses in civility behaviors did not occur in non-civility positive behaviors. Here, instead of highlighting traits that were high HU–low HN, the traits for low HU–high HN were more characteristic than were other traits ($p < 0.001$, $d = 0.95$ for high HU–high HN; $p < 0.001$, $d = 1.2$ for high HU–low HN, and $p < 0.001$, $d = 1.09$ for low HU–low HN). Finally, for negative behaviors, traits that were high HU–low HN were less characteristic than were other traits ($p = 0.003$, $d = 0.41$ for high HU–high HN; $p = 0.021$, $d = 0.30$ for low HU–high HN; $p < 0.001$, $d = 1.26$ for low HU–low HN).

Third, to verify whether high HU and low HN are more characteristic of civil behaviors than of non-civility positive behaviors (H3) and less characteristic of uncivil behaviors than of non-civility negative behaviors (H4), we performed several post hoc Bonferroni-corrected tests between types of behavior on each trait. The results showed that traits with high HU–low HN were more characteristic of civil behaviors than for positive behaviors ($p < 0.001$, $d = 1.20$). As expected, there is a greater inference of high HU traits in civil behaviors than in positive ones. No statistically significant differences appeared in the other three types of traits between the behaviors. Finally, confirming H4, the traits for high HU–low HN were less characteristic of uncivil than of negative behaviors ($p < 0.001$, $d = 0.79$). The same pattern appeared for traits that were high HU–high HN for uncivil behaviors and for negative behaviors ($p = 0.037$, $d = 0.46$).

## 4. Discussion

The aim of Study 1 was to test whether the agents of civil behaviors display the attribution of HU traits to a greater extent than the agents of other positive behaviors. We expected the reverse to be the case for uncivil behaviors. The results confirmed our hypothesis, showing that people infer high HU traits more often when observing other people performing civil behaviors than when performing other positive behaviors, and infer high HU traits less often when observing other people performing uncivil behaviors than when performing other negative behaviors.

The results support Haslam's theory [17], which associates civility with the attributes of human uniqueness. Indeed, in comparison with the other positive behaviors, only civil behaviors facilitate the attribution of this type of trait, while they are not associated with uncivil behaviors, nor with behaviors that are not related to civility behaviors, whether positive or negative. Conversely, the perpetrators of uncivil behaviors show a low score in this type of trait. In accordance with Haslam's model [17], and in line with our hypotheses, the observation of uncivil behaviors gives rise to an animalistic dehumanization of those exhibiting the behavior.

Furthermore, our results for civil and uncivil behaviors differed from those obtained for positive and negative behaviors. HU traits are related to a cognitively sophisticated sense of humanness, in which the socialization process in a particular culture plays an important role [24]. Civility clearly reflects the specific cultural learning required by HU traits, whereas our results indicate that positive behaviors are less closely related to HU.

### 4.1. Study 2

Study 1 showed that civil and uncivil behaviors are associated with the attribution of traits of HU and that civility varies from other types of positive and negative behaviors. In Study 2, we investigated the difference between civil and moral behaviors in the attribution of HU traits.

Is there any difference between civility and morality in the attribution of humanness? Insofar as morality serves to regulate cooperation [25] and to suppress or regulate self-interest [26], it can also be confused with civility. In fact, different authors have linked

civil behaviors to morality [27,28]. However, several studies have shown the relevant differences between moral acts and civil acts. For some scholars, morality is based on moral norms, whereas civility is based on conventional norms [29,30]. Moral norms include acts perceived as "objectively obligated," whereas conventional norms follow situation-dependent rules [31]. In this sense, moral norms are considered universal, because they have also been used to proscribe behaviors in other countries and at other times in history, whereas conventional norms are often localized [32,33].

However, although civility and morality have been conceptually differentiated, the literature on dehumanization considers both dimensions to be equally related to the attribution of HU. In one of the early works about the dual model of dehumanization, Haslam [17] (p. 257) posits that *"when UH characteristics are denied to others, they should be seen as lacking in refinement, civility, moral sensibility, and higher cognition"*. In a recent work conducted by Rodríguez-Pérez [34], the relationship between the dimensions of sociability, morality, and competence and the dual model of dehumanization was explored. The authors concluded that, although competence has great power in predicting HU, morality also plays a relevant role. Therefore, according to the scarce previous research, it can be expected that there will be no differences in the attribution of HU traits when observing people performing civil behaviors and moral behaviors.

We developed four hypotheses in this study to confirm a greater inference of high HU–low HN traits than with other traits for civil and moral behaviors (Hypothesis 1–H1), a lower inference of high HU–low HN traits than with other traits for uncivil and immoral behaviors (Hypothesis 2–H2), an equal inference of high HU–low HN traits in civil and moral behaviors but higher than with positive behaviors (Hypothesis 3–H3), and an equal inference of high HU–low HN traits in uncivil and immoral behaviors but lower than with negative behaviors (Hypothesis 4–H4).

*4.2. Method*

4.2.1. Participants

The participants in this study were 328 university students from Spain ($n$ = 51 in civil, $n$ = 51 in uncivil, $n$ = 55 in moral, $n$ = 51 in immoral, $n$ = 60 in positive neutral, and $n$ = 60 in negative neutral conditions). The participants' ages ranged from 18 to 58 years ($M$ = 19.69, $SD$ = 3.65); 253 were female. A sensitivity analysis conducted with G*Power [22] revealed that the sample was sufficient to detect small effects of $f$ = 0.10 (equivalent to partial $\eta^2_p$ = 0.01), assuming an alpha coefficient of 0.01 and a power of 0.99 (mean correlation among repeated measures = 0.79).

4.2.2. Instruments

*The behaviors.* The same two civil behaviors and two uncivil behaviors presented in Study 1 were included. In the moral condition, two moral behaviors were included: "Think about the people who do not cheat on a test even if they have the answers in front of them" and "Think about the people who stand up for a friend when they are being teased or harassed," whereas in the immoral condition, the following behaviors were used: "Think about the people who cheat on their wife/husband/girlfriend/boyfriend" and "Think about the people who bad-mouth a good friend behind their back". A pre-test study was conducted to test differences in the civility and morality of behaviors. The analyses showed that there were differences in civility between civil behaviors and moral behaviors and between uncivil behaviors and immoral behaviors. Civic and uncivil behaviors were more related to civility than moral and immoral behaviors, respectively. The same analyses were conducted for morality, again obtaining differences between civil behaviors and moral behaviors and between uncivil behaviors and immoral behaviors. Moral and immoral behaviors were more closely related to morality than civic and uncivil behaviors, respectively (see the Supplementary Materials for details of the analysis).

*The traits.* The same traits that were presented in Study 1 were included.

### 4.2.3. Procedure

Following Study 1, in the classroom, the participants completed one of six versions of a paper-and-pencil questionnaire in which they were asked to form an impression of the type of people who exhibit certain behaviors and then respond to a list of traits. Each questionnaire contained a description of two behaviors. After reading each description, the participants were given a list of 32 traits. Specifically, they were asked to score each trait in accordance with the image that they had formed of the person exhibiting the behavior, on a scale from 1 (*The trait does not describe at all the type of person I imagined*) to 5 (*The trait fully describes the type of person I had imagined*).

### 4.2.4. Data Analysis

IBM SPSS Statistics (Version 25) was used for the analyses. A significance level of 0.05 was set. After carrying out a $3 \times 2 \times 2 \times 2$ ANOVA, we conducted several post hoc Bonferroni-corrected tests with the four types of traits derived from the crossing of high versus low values in HU and high versus low values in HN. First, we compared the scores on the four traits in each behavior to test H1 and H2. Then we compared the four behaviors in each trait to test H3 and H4.

### *4.3. Results*

To verify the hypotheses, a 3 (type of behavior: civility vs. moral vs. neutral) $\times$ 2 (valence of behavior: positive vs. negative) $\times$ 2 (HN traits: high vs. low) $\times$ 2 (HU traits: high vs. low) ANOVA was carried out, with the type of behavior and valence of behavior as between-subjects variables and HN and HU as within-subjects variables. The differences in means of the traits that scored high and low in HN and HU for each type of behavior are shown in Supplementary Table S6.

The ANOVA showed that the four-way interaction was statistically significant, $F(2, 322) = 7.10$, $p < 0.001$, $\eta^2_p = 0.42$ (for details, see the ANOVA results summarized in Supplementary Table S7). Several post hoc Bonferroni-corrected tests were carried out between traits on each behavior to verify if both the civil behavior agents and the moral behavior agents were attributed more to high HU–low HN traits than other traits (H1). The results confirmed what we had expected in H1 in civil behaviors but not in moral behaviors. Specifically, for civil behaviors, the traits for high HU–low HN were more characteristic than the other three traits ($p < 0.001$, $d = 0.51$ for high HU–high HN; $p = 0.045$, $d = 0.27$ for low HU–high HN; and $p < 0.001$, $d = 0.84$ for low HU–low HN).

However, a different pattern was found for moral behaviors, in which there was no difference in the attribution of the traits for high HU–low HN and traits for high HU–high HN ($p = 0.646$, $d = 0.05$) and for the traits for low HU–high HN ($p = 0.205$, $d = 0.16$). This is in contrast to civil behaviors because the participants considered all categories of traits to be equally characteristic, except those for low HU–low HN ($p < 0.001$, $d = 0.58$).

For uncivil and immoral behaviors, the post hoc Bonferroni-corrected tests between traits on each behavior confirmed a lower inference of high HU–low HN traits than other traits for both uncivil and immoral behaviors (H2). Specifically, we found that for uncivil behaviors, the traits for high HU–low HN were considered less characteristic than the other three traits ($p = 0.043$, $d = 0.21$ for high HU–high HN; $p < 0.001$, $d = 0.60$ for low HU–high HN; and $p < 0.001$, $d = 1.56$ for low HU–low HN). Additionally, in terms of immoral behaviors, traits that were high HU–low HN were considered less characteristic than the other three traits ($p < 0.001$, $d = 1.32$ for high HU–high HN; $p < 0.001$, $d = 1.73$ for low HU–high HN; and $p < 0.001$, $d = 0.68$ for low HU–low HN).

In contrast to civil and moral behaviors, in positive behaviors, the traits for low HU–high HN were more characteristic than the other traits ($p < 0.001$, $d = 0.92$ for high HU–high HN; $p < 0.001$, $d = 0.95$ for high HU–low HN; and $p < 0.001$, $d = 0.81$ for low HU–low HN). Finally, for negative behaviors, the traits for high HU–low HN were considered less characteristic than the other three traits ($p < 0.001$, $d = 0.79$ for high HU–high HN; $p = 0.030$, $d = 0.32$ for low HU–high HN; and $p < 0.001$, $d = 1.34$ for low HU–low HN).

For H3, we expected to verify whether high HU–low HN traits were considered equal in civil and moral behaviors but higher than in positive behaviors. The post hoc Bonferroni-corrected tests between the types of behavior showed no statistical differences between traits for civil and moral behaviors ($p = 0.072$, $d = 0.31$) in the traits for high HU–low HN. However, these traits were less commonly attributed to positive behaviors ($p < 0.001$, $d = 1.09$ for civil and $p < 0.001$, $d = 0.67$ for moral). Furthermore, the traits for high HU–high HN were considered less characteristic for positive than for civil ($p = 0.012$, $d = 0.55$) and moral behaviors ($p < 0.001$, $d = 0.66$).

Finally, in H4, we expected to verify an equal attribution of HU in uncivil and immoral behaviors but lower than in negative behaviors. In line with our hypothesis, the post hoc Bonferroni-corrected tests between behaviors showed no statistical differences between traits for uncivil and immoral behaviors ($p = 0.056$, $d = 0.37$) in the traits of high HU–low HN. However, these traits were considered more characteristic for negative behaviors than for uncivil ($p < 0.001$, $d = 0.99$) and immoral behaviors ($p = 0.001$, $d = 0.80$).

The traits for high HU–high HN were considered less characteristic for uncivil than for immoral behaviors ($p < 0.001$, $d = 1.13$) and negative behaviors ($p < 0.001$, $d = 1.29$). A different pattern was found for the traits for low HU–low HN, with lower scores for immoral behaviors than for the other behaviors ($p = 0.002$, $d = 0.60$ for uncivil behaviors and $p < 0.001$, $d = 0.96$ for negative behaviors). Moreover, the traits for low HU–high HN were considered more characteristic for immoral behaviors ($M = 2.92$) than for uncivil ($p < 0.001$, $d = 1.15$) and negative behaviors ($p < 0.001$, $d = 0.71$).

*4.4. Discussion*

The aim of Study 2 was to test whether the agents of civil and moral behaviors displayed the inference of HU traits to the same extent. The results confirmed those that we obtained in Study 1 for civil and uncivil behaviors: The inference of HU traits is higher when presenting civil behaviors compared with other positive behaviors.

However, our results showed that civil and uncivil behaviors do not have an exact correspondence with moral and immoral behaviors. Whereas, for civil behaviors, the HU traits were attributed to a greater extent than were the other traits, a different pattern was found for moral behaviors, in which there was no difference in the attribution of traits for high HU–low HN, traits for high HU–high HN, and traits for low HU–high HN. Morality seems to be a complex facet of humanness because it is related not only to HU traits but also to HN traits. Previous research [19] has related the HN traits to moral acts, pointing out that the desire to actively engage in moral behavior (proactive agency) is related to warmth and the emotional characteristics of HN traits.

In fact, the results of our study also indicate that immoral behaviors promote the inference of HN traits. Specifically, the traits for high HN–low HU were considered more characteristic of immoral behaviors than of uncivil and neutral behaviors. It would make sense to relate HN to morality when the theoretical difference between civility and morality is based on the universal character of the latter. Future studies could help elucidate the difference in HN traits between morality and civility and how this relates to the universality or specificity of culture.

## 5. General Discussion

Despite the theoretical relevance of the concept of civility in the study of dehumanization, little research has been conducted into civil behavior and how it is associated with humanness. Across two studies, our results showed a consistent association between civil behaviors and humanness; specifically, civil behaviors lead to the inference of HU traits. Additionally, our results revealed that civil and uncivil behaviors display a different pattern of associations with human traits than moral and immoral behaviors.

The data from the first study confirmed that civil behaviors display a differential attribution of HU traits compared with other positive behaviors, which is congruent with Haslam's theory [17]. Furthermore, we observed that uncivil behaviors constitute an

obstacle to the inference of HU traits to a greater extent than any other negative behavior not related to civility. In this sense, our data confirm that uncivil behaviors promote the animalistic dehumanization of those exhibiting this type of behavior.

The second study extends the hypothesis of a link between civility and uniquely human traits to moral behaviors. Our results showed that agents of civil and moral behaviors display a different pattern in terms of humanness. Specifically, morality is related not only to HU traits but also to HN traits. An explanation could be related to the link between universality and moral norms. In this sense, moral norms are considered "objectively obligated", that is, they are common to other countries and other times, and therefore are unlike the norms of civility that represent conventional norms that are determined locally by the concrete learning of a culture; moral standards are universal [29–32]. For their part, Haslam et al. [35] state that HU is related to enculturated humanness, while HN corresponds to common humanness. That is, HU is related to what is culturally learned, and HN to what is universal and characteristic of the human being. In this sense, considering the literature that associates morality with universal norms and civility with conventional norms [29], one could argue that morality should be associated with HN to a greater degree than civility. However, the dual model of dehumanization does not suggest this relationship [17]. Despite this, Bastian et al. [19] verified that moral status is associated in distinctive ways with the two dimensions of humanness. While aspects of moral status, such as the inhibiting agency (i.e., responsibility for immoral behavior) are related to HU, others such as the pro-active agency (i.e., the capacity to engage in moral behavior) and moral patiency (i.e., the capacity to be recipients of morally relevant actions) relate to HN.

This research has theoretical implications for dehumanization theory. First, our results constitute an advance in the understanding of relationships between civility and humanness. To date, no empirical studies have explored the social perception of incivility framed within dehumanization theory. Previous studies have provided evidence that moral judgments and dehumanization are closely connected [e.g., 19]. Indeed, immoral actions, even if they are distinctively human, such as torture, clearly have dehumanizing outcomes for the perpetrators [20]. A recent study [21] revealed an automatic association between uncivil behaviors and humanness, suggesting a possible link between this association and the social acceptance of these types of behaviors when they are framed as typically human actions. In our view, the results of this research represent an advance that empirically confirms the theoretical link between uncivil behavior and uniquely human perception.

Research that leads to a deeper understanding of the social perception of perpetrators of uncivil and immoral behaviors should compare and differentiate them. Studies that help toward a better understanding of the links between civility, morality, and humanness could clarify several outstanding questions in dehumanization theory. To date, the differences between morality and civility have been presented theoretically but have not been sufficiently explored empirically. Recently, despite the substantial body of research and theory on dehumanization that has been developed over the last two decades, the explanatory power of dehumanization theory has been questioned [36]. To refute this position, experts in the field have highlighted that humanness and moral evaluation are two related but distinct processes [37]. Future studies would help us better understand the links and differences between the inferences drawn for each type of behavior. In this sense, the analysis of civility and its consideration as a "partially restrictive" or "hierarchically restrictive" dimension [38,39] would be interesting. Hierarchically restrictive traits are concerned with morality or ability, but no studies have shown whether civility is associated with hierarchically restrictive or partially restrictive traits. If a moral person performs an immoral act, the impressions of that person change to considering them immoral, but if an immoral person performs a moral act, impressions of the person are not changed [38,40]. What can one expect of a civil person? Does a single behavior that contradicts this impression lead to changing the impression of that person to an uncivil person? If an uncivil person performs a civil behavior, do the impressions of others regarding that person change? Future studies

will help shed light on these and other issues of interest to differentiate between moral and civil acts.

The results we obtained should be considered with caution, given that there are several limitations to the studies presented here. First, the studies worked with only two behaviors of each type (civil/uncivil/neutral positive/neutral negative). The list of traits that was used to evaluate the target was long and placed a mental burden on the participant; therefore, only two items were used to ensure that the participants did not complete the task randomly due to fatigue. This could lead to results that are closely linked to the behaviors presented. To generalize the results, it would be necessary to obtain consistent results in additional studies with a broader range of behaviors because there are different kinds of moral violations—some are due to an excess of "animal passion" (such as lust or anger), while others are due to alienation from feelings (such as cruelty). It is quite possible that the results may depend on which kind of immorality is most salient. Moreover, different thematic behaviors were used. Employing the same thematic contents for civil and uncivil behaviors would have allowed for testing the effect of civil versus uncivil behaviors on the dependent variables while controlling for the effects of additional characteristics of the stimuli. Future studies could address this issue. Furthermore, the sample is not balanced by gender due to convenience sampling. Future studies could consider whether the results are mediated by the gender of the evaluator and also by the gender of the perpetrator who performs the uncivil behavior. In addition, an intercultural perspective must be considered to account for the variability in behavioral norms and patterns. It would also be interesting to carry out these studies with different samples to determine whether the participant's sex or age might lead to differences in evaluating civil behaviors. Finally, another interesting future line of research would be to study how the traits of HU are inferred when the same person performs both civil and uncivil behaviors.

## 6. Conclusions

In conclusion, the results of our studies corroborate the theoretical proposal that civility is a central dimension of one of the types of humanness that Haslam [17] proposed, that of HU. According to our results, observing others exhibiting civil behaviors facilitates the inference of HU traits. Importantly, the agents of civil and moral behaviors are differentially perceived in terms of humanness. Civil and moral behaviors, therefore, constitute a way to explore the attribution of humanness in interpersonal and intergroup relations.

**Supplementary Materials:** The following supporting information can be downloaded at: https://www.mdpi.com/article/10.3390/ejihpe12090098/s1, Pretest study: Pretest study of 120 civil and uncivil behaviors (Study 1), Table S1: Sample size be-fore and after correlation analysis and Cronbach's alpha (Pretest Study 1), Table S2: Mean scores and standard deviation of 120 civil and uncivil behaviors in eight dimensions related to the per-ception of humanness (Pretest Study 1), Table S3: Means and standard deviation of each of the clusters, and correlations between dimensions (Pretest Study 1),Behaviors selection (Study 1), Traits selection (Study 1), Behaviors selection (Study 2), Table S4: Descriptive Statistics for Positive and Negative Behaviors Ratings in Humanity Traits in Study 1, Table S5: F-ratios resulting from the repeated-measures ANOVA (Study 1), Table S6: Descriptive Statistics for Positive and Nega-tive Behavior Ratings in Humanity Traits in Study 2, Table S7: F-ratios resulting from the repeat-ed-measures ANOVA (Study 2).

**Author Contributions:** Conceptualization, L.R.-G., N.D., A.R.-P., R.R.-T. and V.B.; data curation, L.R.-G.; formal analysis, L.R.-G., N.D., A.R.-P. and R.R.-T.; funding acquisition, A.R.-P. and V.B.; investigation, A.R.-P.; methodology, L.R.-G., N.D., A.R.-P. and R.R.-T.; project administration, A.R.-P. and V.B.; resources, A.R.-P. and V.B.; software, L.R.-G., A.R.-P. and R.R.-T.; supervision, L.R.-G. and N.D.; validation, L.R.-G.; visualization, L.R.-G. and N.D.; writing—original draft, L.R.-G., N.D., A.R.-P., R.R.-T. and V.B.; writing—review and editing, L.R.-G. and N.D.. All authors have read and agreed to the published version of the manuscript.

**Funding:** This research was funded by the Spanish Ministry of Economy, Industry, and Competi-tiveness, grant number PSI2016-78450-P and Spanish Ministry of Science and Innovation: PID2019-108217GB-I00.

**Institutional Review Board Statement:** The study was conducted according to the guidelines of the Declaration of Helsinki and approved by the Ethics Committee on Research and Animal Welfare of the University of La Laguna (CEIBA2021-0442).

**Informed Consent Statement:** Informed consent was obtained from all subjects involved in the study.

**Data Availability Statement:** The datasets generated for this study are available at https://osf.io/jfg6a/?view_only=6a1e6b210338480e9eb383d191c80a76 (accessed on 12 September 2022).

**Conflicts of Interest:** The authors declare no conflict of interest. The funders had no role in the design of the study; in the collection, analyses, or interpretation of data; in the writing of the manuscript, or in the decision to publish the results.

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
