# Peer review of "Dehumanization as a Response to Uncivil and Immoral Behaviors"

_ejihpe, doi:10.3390/ejihpe12090098_

Round 1

Author Response

Response to Reviewer 1 Comments

The paper deals with the framework of dehumanization and explores the link between civility and humanness. In the first study, the authors verify whether civility relates to one of the specific components of humanness, human uniqueness. In the second study, they extend the analysis of the link between civility and human uniqueness to morality, which constitutes a theoretical advancement in understanding the constructs of civility and humanness. The hypotheses are clearly formulated, the study's design and the data analyses are correct, and recent literature is appropriately quoted in the introduction. Overall, the paper is well-written, and the number of participants and the research methodology is adequate. However, after my reading, I would suggest a few improvements, as follows:

Point 1: On line 53, The authors state that “Most of the measures of dehumanization have indirectly shown a possible link between civility and humanness. For example, Implicit Association Tests (IATs) have illustrated that people unconsciously associate more quickly and accurately ingroup (vs. outgroup) names with human-related (e.g., inhabitant, citizen) versus animal-related (e.g., irrational) words”. However, I do not understand how this example shows a link between civility and humanness. In my opinion, this is unnecessary, or the authors should address in more detail this example.

Response 1: We agree with the reviewer. In order to gain in clarity, we decided remove this information.

Point 2: Briefly describe how civil, uncivil, and unrelated to civility behaviors were selected in the pre-test.

Response 2: Following this suggestion, in the new version of the manuscript we have added information from the pre-test on the choice of each type of comment.

Point 3: Describe how the moral and immoral behaviors used in study 2 were selected.

Response 3: We clarify the selection of each type of behavior used in the study 2.

Point 4: The sample is composed of college students and is not balanced by gender due to convenience sampling. Anyway, I wonder if the gender variable can interact with the retrieved effect.

Response 4: We agreed and decided to include this limitation in the limitation and future reseach section.

Point 5: On line 402 authors state, “An explanation could be related with the link between universality and moral norms.” The authors should address these issues more in detail to justify the conclusions better.

Response 5: Thank you for your comment. We have developed this idea to give more clarity to the paragraph.

Thank you very much for your comments and suggestions, we hope that the new version of the manuscript would clarify the aspects found in the previous version.

Reviewer 2 Report

This is a well-conducted study examining the link between civility and human uniqueness. Two studies were conducted. The findings were interpreted to support the argument that civil behaviors facilitate the inference of humanness, and civil and moral behaviors are not equivalent in terms of the human inferences they lead to.

Overall, the paper is well-written. The sample size is adequate. I also impress by the fact that the authors shared the data and material at OSF. I believe that the paper will be able to contribute well to the literature. In fact, the paper is ready for publication. I only have one minor comments to improve the manuscript further:

1. I feel that it will be helpful if the authors can elaborate more on their pre-test studies in the main text. The information is useful and might be wasted in the supplementary materials.

Author Response

Response to Reviewer 2 Comments

This is a well-conducted study examining the link between civility and human uniqueness. Two studies were conducted. The findings were interpreted to support the argument that civil behaviors facilitate the inference of humanness, and civil and moral behaviors are not equivalent in terms of the human inferences they lead to.

Overall, the paper is well-written. The sample size is adequate. I also impress by the fact that the authors shared the data and material at OSF. I believe that the paper will be able to contribute well to the literature. In fact, the paper is ready for publication. I only have one minor comments to improve the manuscript further:

Point 1: I feel that it will be helpful if the authors can elaborate more on their pre-test studies in the main text. The information is useful and might be wasted in the supplementary materials.

Response 1: We agree with the reviewer. In order to gain in clarity, we decided to add information about pre-tests in the main text.

Thank you very much for your comments and suggestion, we hope that the new version of the manuscript would clarify the aspect found in the previous version.
